# Challenges for the Routine Application of Drones in Healthcare: A Scoping Review

Sara De Silvestri [1], Pasquale Junior Capasso [2], Alessandra Gargiulo [1], Sara Molinari [2] and Alberto Sanna [1,*]

1   Center for Advanced Technology in Health and Wellbeing, IRCCS Ospedale San Raffaele, Via Olgettina, 60, 20132 Milano, Italy; desilvestri.sara@hsr.it (S.D.S.); gargiulo.alessandra@hsr.it (A.G.)
2   EuroUSC Italia, Via Daniele Manin, 53, 00185 Roma, Italy; pasqualej.capasso@eurousc-italia.it (P.J.C.); sara.molinari@eurousc-italia.it (S.M.)
*   Correspondence: sanna.alberto@hsr.it

**Abstract:** Uncrewed aerial vehicles (UAVs), commonly known as drones, have emerged as transformative tools in the healthcare sector, offering the potential to revolutionize medical logistics, emergency response, and patient care. This scoping review provides a comprehensive exploration of the diverse applications of drones in healthcare, addressing critical gaps in existing literature. While previous reviews have primarily focused on specific facets of drone technology within the medical field, this study offers a holistic perspective, encompassing a wide range of potential healthcare applications. The review categorizes and analyzes the literature according to key domains, including the transport of biomedical goods, automated external defibrillator (AED) delivery, healthcare logistics, air ambulance services, and various other medical applications. It also examines public acceptance and the regulatory framework surrounding medical drone services. Despite advancements, critical knowledge gaps persist, particularly in understanding the intricate interplay between technological challenges, the existing regulatory framework, and societal acceptance. This review highlights the need for the extensive validation of cost-effective business cases, the development of control techniques that can address time and resource savings within the constraints of real-life scenarios, the design of crash-protected containers, and the establishment of corresponding tests and standards to demonstrate their conformity.

**Keywords:** uncrewed aerial vehicle; unmanned aircraft system; medical drone; healthcare logistics; AED drone delivery; scoping review; drones in public health; drone transport



## 1. Introduction

Uncrewed aerial vehicles (UAVs), commonly called drones, are aircraft that operate autonomously or are piloted remotely without a pilot on board. In recent years, their utilization has expanded, particularly in the medical sector. Drones offer the advantage of the swift transportation of medical materials, proving critical in time-sensitive situations. They can play a pivotal role in reaching remote or challenging-to-access areas, facilitating the delivery of medical services to underserved populations. Additionally, drones contribute to emergency response efforts and disaster situations, especially in areas with compromised road infrastructure. Beyond their life-saving potential, drones have been studied to offer economic benefits by realizing long-term cost efficiency and minimizing facility costs due to their minimal spatial requirements for take-off, landing, and storage.

Time and cost efficiency play crucial roles in the consideration of drone utilization in healthcare. The exploration of drones in various healthcare innovations has been documented in prior literature reviews. For example, they have been used for the transport of automatic external defibrillators (AEDs) in out-of-hospital cardiac arrest (OHCA) events in a reduced amount of time [1]. Indeed, drones can successfully deliver AEDs and, in most cases, they arrive before ambulances [2]. This timely intervention reduces the crucial

time needed for the first defibrillation, ultimately contributing to saving lives. Another time-sensitive application is organ delivery, which can be accomplished by a drone directly traveling from the donor to the recipient hospital, ensuring the organ's proper condition throughout the flight. [3]. Drone transport can also significantly improve the delivery time of blood and blood products. Research has shown that drones can efficiently transport blood products, thereby saving lives in emergency situations and minimizing wastage in healthcare facilities [4]. Ensuring temperature control and preserving the physical integrity of perishable goods during transport, such as blood samples, is crucial for medical deliveries. This is particularly important as temperature alterations and vibrations can affect the quality of these sensitive medical materials [5]. In a 2020 literature review, several studies highlighted the potential of drones in vaccine transport by increasing the availability and decreasing the operational costs [6]. A scoping review delved into human–drone interaction in medical supply delivery, exploring people's responses to these innovative transportation methods [7]. Concurrently, some reviews focused on specific drone aspects or applications, such as surveillance in disaster areas and epidemiological monitoring [6,8]. Additionally, one review is dedicated to emergency medical services [9].

Selecting the optimal solution for delivering medical services is crucial, especially in emergencies in which the delivery time is paramount. For this reason, different solutions to support the human decision in goods transportation have been developed [10,11]. Moreover, Laksham [12] conducted a SWOT (Strengths, Weaknesses, Opportunities, and Threat) analysis to evaluate the likelihood of the success or failure of drone applications in public health. The analysis highlights the significant opportunities drones offer in efficiently transporting blood, specimens, vaccines, and other biologicals to remote locations, thereby reducing the travel time and saving lives. However, there are weaknesses and threats that research and technological advancements can tackle. Public acceptance emerges as a potential hurdle to drone implementation in healthcare. Rejeb et al. [13] identified three main barriers to the implementation of humanitarian drones: technological, organizational, and environmental. Addressing user acceptance, a specific challenge within the organizational barrier, can be addressed through technology adoption theories, use case studies, interviews, and surveys.

The objective of the present study is to address the gap in existing reviews by conducting a scoping review on the use of drones in healthcare. The primary rationale is to gain a comprehensive understanding of the applications of drones in the medical field, encompassing their advantages, disadvantages, and the key challenges associated with their application in healthcare. Additionally, this review aims to provide insights into the current state of drone research in healthcare, identifying technological and societal gaps that require attention in future research.

The remainder of the paper is organized as follows: Section 2 describes the material and methods utilized for this scoping review, how the articles have been selected and classified. In Section 3, the results are presented and analyzed according to their respective categories, while they are further discussed and interpreted in Section 4, in which the limitations of this review are addressed and the directions of future research are highlighted.

## 2. Materials and Methods

The protocol of this scoping review was developed a priori, in conformity with the Preferred Reporting Items Systematic Reviews and Meta-Analyses Extension for Scoping Reviews (PRISMA-ScR) [14], in accordance with the guidance developed by the Joanna Briggs Institute [15]. Therefore, the search criteria, article selection, and subsequent information to extract from the papers were priorly defined. The related protocol is explained in the following sections.

### 2.1. Search Methods and Article Selection

The search criteria were intentionally broad to encompass all of the relevant scientific evidence of the application of drones in healthcare within the timeframe of interest, span-

ning from 1 January 2021 to 7 March 2023. A specific search period was defined because this review's aim was to analyze the gaps in the scientific research within a rapidly evolving field. The beginning of 2021 was set as the start date, as it aligns with the applicability start date of the Commission Implementing Regulation (EU) 2019/947, which governs European UAS operations.

Peer-reviewed journal papers were included only if their primary focus was the use of drones in healthcare and if written in English. Since the research question regards specifically the scientific literature published on the subject, grey literature was not included. Quantitative, qualitative, and mixed-methods approach studies were included to map the different methodologies adopted by studies that were produced on the subject.

A systematic literature search was conducted using the following databases: PubMed, ScienceDirect, Scopus, and EMBASE. The authors of this review individually drafted search strategies, which were later consolidated through a collective discussion. The search terms used with each database were classified at two levels: (1) "healthcare", "hospital *", "medic *", "pharmaceutic *", "diagnostic sample *", "drug *", and "defibrillator *", AND (2) "drone *", "Unmanned aerial vehicle *", "Unmanned aircraft *", "Urban air mobility", "Advanced air mobility", "Innovative air mobility", "U-space", and "Unmanned traffic management". All combinations between the two levels of search terms were used in all of the mentioned databases.

All papers retrieved with the proposed search strategy were screened depending on the broadly defined inclusion criteria. Specifically, the focus of the selected papers was the design, development, or evaluation of the healthcare services provided through the use of drones. As an example, journal papers that focused on search-and-rescue operations were not included if the drone's intervention was aimed at locating missing individuals. However, they satisfied the inclusion criteria if the drone provided any kind of support to triage, diagnosis, or treatment before the hospitalization of the rescued individuals. Likewise, studies were excluded if the drones were merely provided as one among multiple examples of the application of a specific technology (e.g., algorithms, blockchain, and 5G networks).

All references of the articles found with the described search strategy were uploaded on the webapp Rayyan [16]. After the software-aided removal of duplicates, each article's title, abstract, keywords, and metadata were screened independently by two of three authors (S. D. S., P. J. C., A. G.) to decrease the reviewers' bias. Each reviewer expressed a blind vote on the inclusion of two-thirds of the articles. Those with unmatching votes or marked as "undecided" by one reviewer were discussed collectively and resolved by reaching consensus.

Included articles were independently analyzed in full text by three reviewers (P. C., A. G., S. M.) to further assess each paper's eligibility based on the aforementioned criteria.

### 2.2. Data Extraction

Data extraction was conducted systematically by the same reviewers who performed the full-text screening. Data charting elements comprised the following: authors, research type, focus domain, objective, methodology, key findings, and gaps in research or implementation. The papers were classified by the focus domain and the methodology applied, and a qualitative summary of the findings divided by the focus domain was conducted.

## 3. Results

### 3.1. Study Selection

The number of records identified by the initial search amounted to 1178. Of those, 419 were removed by the semi-automatic duplicate removal tool. A screening of the title, abstract, keywords, and article metadata was performed on 761 records, of which only 119 conformed to the previously defined eligibility criteria and were, hence, sought for retrieval (n = 99/119). Thirty articles that were analyzed in full text were excluded for the following reasons: reviews (n = 4), primary focus different from the use of drones in healthcare

(n = 21), not peer-reviewed (n = 3), and retracted (n = 2). Indeed, during the article screening phase, two articles were retracted by the journal because the accuracy or validity of the results were deemed questionable after publication. Finally, 69 articles passed selection and were used for the current review. Figure 1 shows a flow diagram of the selection process following the PRISMA-ScR guidelines.

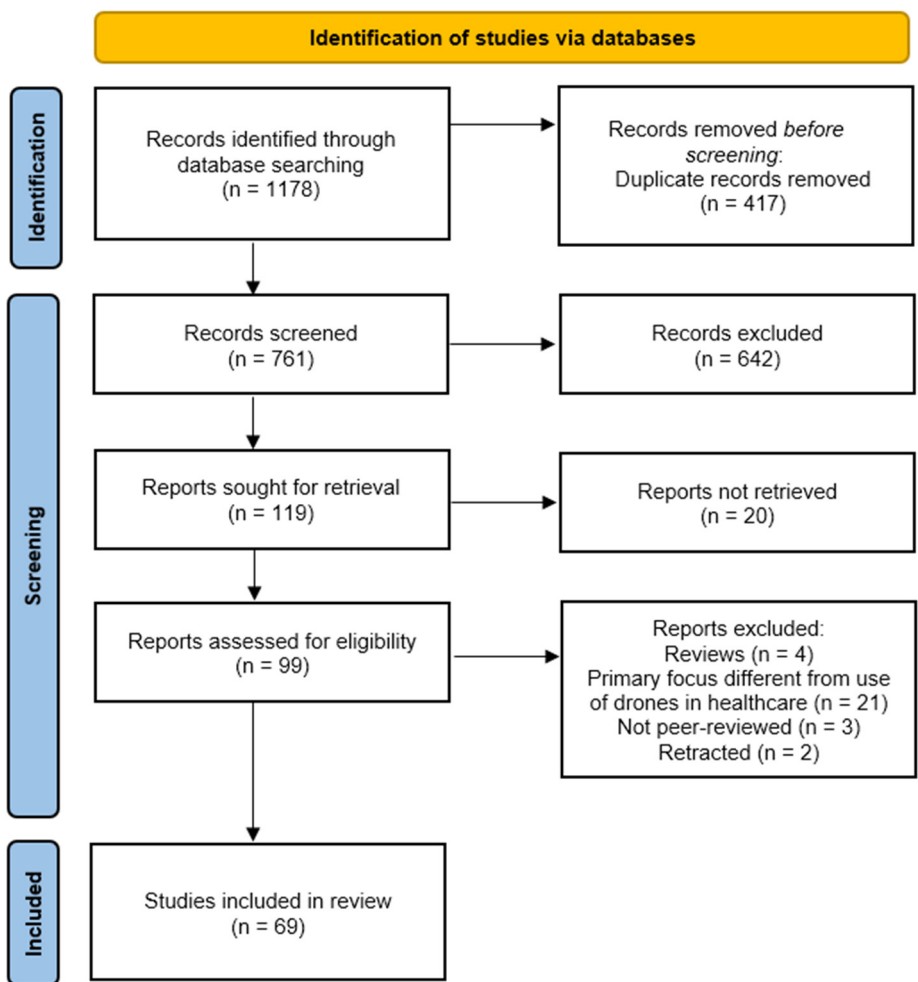

**Figure 1.** Prisma 2020 flow diagram for systematic reviews [17].

*3.2. Study Characteristics*

The analysis of the eligible papers led to the identification of the following focus domains: transport of biomedical goods; AED delivery; healthcare logistics; air ambulance; other medical applications; public acceptance of medical drone services; and regulatory framework concerning medical drone services.

The papers were categorized by the research methods applied: empirical drone research (n = 21); prototype design (n = 9); retrospective data analysis (n = 6); computational simulation (n = 20); social research (n = 7); and theoretical modeling (n = 5). The studies under review were categorized based on both their focus domains and research types. Table 1 provides a visual representation of the fields that were more frequently explored in the specified timeframe, along with the various study designs employed.

**Table 1.** Mapping of reviewed papers based on research type and study domain.

|  | Empirical Drone Research | Prototype Design | Retrospective Data Analysis | Computational Simulation | Social Research | Theoretical Modeling |
|---|---|---|---|---|---|---|
| Transport of biomedical goods | [18–25] | [26,27] | [28] | [29–32] | [33–35] | [36,37] |
| AED delivery | [38,39] | - | [40,41] | [42–52] | - | [53] |
| Healthcare logistics | [54–62] | [63–68] | [69] | [70] | [71,72] | [73] |
| Air ambulance | - | [74] | [75] | [76] | - | - |
| Other medical applications | [77,78] | - | - | [79–81] | - | [82] |
| Public acceptance | - | - | - | - | [83,84] | - |
| Regulatory framework | - | - | [85] | - | - | - |

*3.3. Qualitative Summary and Synthesis*

3.3.1. Transport of Biomedical Goods

The transportation of critical biomedical goods using drones represents a unique solution to overcome transportation challenges in remote or inaccessible areas, as well as during emergencies or natural disasters, helping to enhance patient care and save lives.

A total of 20 papers focused on the transport of biomedical goods: 6 (30%) dealt with the transport of blood, 4 (20%) analyzed the transport of medicines, 4 (20%) explored the transport of biological samples, while 4 (20%) focused on strategies to optimize the use of UASs for medical supplies in emergency situations. Furthermore, two (10%) papers were about the transportation of human organs, showing how drones have recently become attractive for the delivery of such critical items. According to a survey addressed to transplant surgeons [35], the method of transportation was deemed insignificant by 90.9% of the participants when deciding whether to accept an organ, demonstrating a readiness to consider the use of UAVs for this specific objective. The findings gained from this questionnaire revealed a dichotomy in participants' sentiments towards drones. While a noteworthy 34.5% of the respondents admitted to experiencing apprehension in the proximity of drones, and 23% acknowledged being genuinely alarmed by their utilization, the prevailing consensus among the respondents was a belief in the potential of drones to provide assistance to individuals. Furthermore, a substantial 72.7% of the participants expressed a conviction that UAVs can play a pivotal role within the field of medicine. In 2019, a team of multidisciplinary experts developed, constructed, and tested the first drone system specifically designed for organ transportation. Through this system, the first successful delivery of a human organ using unmanned aircraft was accomplished in Maryland, USA [27].

The use of UAVs for the transportation of biomedical items offers numerous advantages. First, the potential for a reduction in delivery times has been recognized by several studies [20–22,28], stressing the efficacy of drone utilization as an alternative means of transportation for medical purposes when compared to conventional modes. A retrospective data analysis was conducted to assess the impact of UAVs on the delivery of blood products and associated wastage in Rwanda [28], where the government launched a drone delivery program for blood products in 2016, spanning 20 healthcare facilities across the country. The research employed two distinct methodologies: first, a cross-sectional comparison of delivery times between drones and traditional road-based delivery methods, with data collected from March 2017 to December 2019; secondly, an interrupted time series analysis to evaluate changes in the expiration of blood components before and after the implementation of the drone delivery program. The findings revealed that the utilization of drones in Rwanda resulted in significantly shorter delivery times, with deliveries occurring 79 to 98 min earlier than traditional road transportation. Furthermore, the introduction

of the drone delivery program was associated with a noteworthy 67% reduction in the expiration of blood and blood products.

Time efficiency and feasibility were also demonstrated in the simulation study conducted within the Himachal Pradesh region of India [21], wherein a total of 151 transfers of sputum samples were executed using both UAVs and the conventional motorbike transport system. In addition to highlighting the reduction in the median delivery time achieved through UAVs (6.9 min, as opposed to 21.9 min via the traditional system), this investigation substantiates the cost-effectiveness of the proposed method. Indeed, this study stands as the sole research among those under review to present a feasibility study demonstrating a significant reduction in recurrent transportation costs, with expenses as low as INR 20 for UAVs, in contrast to INR 85 associated with the traditional motorcycle-based system.

Moreover, through the persistent optimization of algorithms focusing on routing and path planning issues, researchers proposed theoretical models for achieving higher efficiency and decreasing energy consumptions in drone systems [19,23,29,30,36]. A research study by Shi et al. [36] proposed a bi-objective mixed-integer programming model for the multitrip drone location routing problem, which allows for simultaneous pickup and delivery. This optimization model, beyond shortening the delivery time, was demonstrated to be safer and able to save more resources compared with separate pickup and delivery modes.

Because of the sensitivity and critical nature of biomedical goods, part of the existing literature has focused on ensuring optimal transportation conditions. In this regard, two experimental studies have delved into the impact of drone vibrations on the integrity of transported materials. Specifically, Oakey et al. [24] conducted research aimed at quantifying the vibrations experienced during drone flights and examining their potential effects on the quality of medical insulin through live flight trials. Although the quality of insulin remained unaffected, it was observed that the overall vibration levels were significantly higher compared to traditional road transportation. This finding highlights the need for further in-depth exploration in this area. Similarly, Zhu et al. [25] investigated the influence of drone flights on the stability of cancer medicines. Again, it was determined that the vibration levels induced by drone flights were not significant enough to adversely impact the quality of these medications.

Also, the impact of different materials used in drone carriage on the maintenance of the optimal storage temperature has been investigated. In a simulation study conducted in Malaysia [32], different materials were compared based on their quality in preserving the desired mean kinetic temperature for transporting blood in an equatorial climate; as a result, EPS foam molded into a six-faced cuboid shape proved to be the most effective. The study found that the blood quality remained stable, with no significant differences observed when compared to the control samples. In contrast, Gan et al. [26] developed an application-based temperature monitoring system specifically designed for medical deliveries. Unlike traditional systems used in blood transportation, which only allow for temperature data retrieval through a USB cable after the delivery is completed, their system incorporates both active and passive cooling mechanisms to ensure a consistently cool environment during the flight mission. Moreover, this system enables the real-time monitoring of temperature and humidity, thereby ensuring the preservation of a blood sample's quality.

However, a series of limitations in using drones to transfer biomedical goods have been identified. The foremost limitations frequently cited include a significant dependence on weather conditions [19–21] and technical constraints, such as battery lifespan and transport capacity [22,26,31,37]. Adverse meteorological conditions, such as strong winds or rainfall, hinder the deployment of unmanned aerial vehicles, thus providing an intermittently unavailable service. Additionally, technical issues impede the full potential of drone services; for example, a limited battery life restricts the achievable delivery distance, making long-distance deliveries unfeasible without scheduled stops for battery replacement or recharging.

Furthermore, the potential beneficial impact of employing UAVs is severely limited by the absence of adequate regulations tailored to drone delivery operations in Europe [18,33] and Pakistan [19]. The studies reviewed consistently highlight the necessity for appropriate regulations, specifically focusing on privacy and noise. This lack of regulatory frameworks is even more concerning when considering the transportation of hazardous biomedical goods. The transportation of such goods via UAVs remains a relatively unexplored domain that has not yet received comprehensive attention from regulatory bodies, governments, industry associations, professionals, or academic institutions [34].

### 3.3.2. AED Delivery

Out-of-hospital cardiac arrest is the third principal cause of mortality in industrialized nations, resulting in approximately 275,000 cardiac arrests annually in Europe [31]. Despite significant progress in this field, the survival rates remain low. Timely defibrillation within 3–5 min after OHCAs can yield survival rates of 50–70% [46], while each minute of delay reduces the probability of survival by 10%. Surprisingly, less than 2% of OHCA victims receive automated external defibrillator assistance before the arrival of emergency medical services. To mitigate delays in AED deployment during OHCA incidents, a series of initiatives have been implemented and drone technology is increasingly regarded as a viable option.

In fact, a total of 16 papers focused on this topic. Numerous simulations and physical experimental studies [39,42,43,47,48,50,53] have conducted comparisons between ground-based AED delivery methods and AED-equipped drones, with findings consistently indicating that drones offer a promising means to enhance response times. For instance, a numerical simulation carried out in Greater Paris [39] revealed that a maximum of 26% of OHCA patients received an AED delivered by a drone prior to the arrival of the basic life support (BLS) team, resulting in a time reduction of 190 s in 93% of cases. Similarly, a separate study conducted in Germany [47] dispatched drones from five drone bases to a simulated OHCA scenario, with 10 flights scheduled per route, achieving drone defibrillation times below the average regional emergency medical services (EMS) time. Notably, the AED-drone delivery times ranged from approximately 6 min to 15.5 min, depending on the distance covered, representing a significant improvement compared to the average EMS defibrillation time of 19.5 min in 2019.

Also, AED-equipped drones have been recognized as valuable in increasing the coverage of emergency medical services, especially in remote or geographically challenging areas [41,46]. In a study conducted in Sweden, the geographical information system (GIS)-analysis was used to identify high incidence areas of OHCAs, emerging as a tool to quantify the need of AED-equipped drones. In fact, researchers found that to reach all OHCAs in high-incidence areas in Sweden with AEDs delivered by drone or ambulance within eight minutes, 61 drone systems would be needed. Thus, by introducing a small number of drones, the overall coverage of OHCAs would increase by 58.2%, with a median time savings of about 5 min [41].

Nevertheless, it is important to note that most of the abovementioned studies [39,42, 43,47,48,50,53] were conducted under ideal conditions, intentionally excluding factors such as adverse weather conditions, drone reliability, and obstacles that could potentially impact drone travel times. Consequently, the actual time benefits associated with AED-equipped drones may have been overestimated.

The only study within the reviewed literature that specifically examines the influence of topographic and weather conditions on UAS-AED operations is a retrospective analysis conducted by Choi et al. [40]. The findings from this study suggest that when accounting for environmental and geographical factors in an urban area with a well-organized emergency medical system (such as the city of Seoul), the time saved through UAV-AED deployment is reduced. Specifically, the AED attachment time was found to be lower than the results reported in previous studies.

Furthermore, AED drones proved to be feasible for deliveries in real-life scenarios. In fact, both computational and physical simulation studies assessed the effective possibility of employing UAVs for AED during daytime [48,51,53] and nighttime [44] deliveries in each specific analyzed area. Specifically, in the feasibility study focusing on nighttime drone operations [44], the comparison of operational and safety data showed no major differences between daytime and nighttime use. Actually, despite the slightly inferior visibility of the target site, night landings were marginally quicker on average than daytime landings.

Nevertheless, the cost effectiveness of AED-carrying drones is still controversial. Various studies mention the expenses associated with their implementation as one of the primary limitations to their practical usability [45], together with the existing regulatory gaps [48,49,52]. However, contrasting findings from cost analyses conducted by Röper et al. [38] and Bauer et al. [52] indicate that an AED system based on UAVs is more cost-efficient than the conventional stationary solution. In particular, Bauer et al. [52] assessed the cost-effectiveness of drones in delivering AEDs in Germany through a location-allocation analysis. The study estimated that an 80% coverage of challenging-to-access areas would necessitate 800 drones, resulting in an additional 1477 life-years saved at an annual cost of approximately EUR 18 million.

### 3.3.3. Healthcare Logistics

The proper functioning of a healthcare system heavily relies on an efficient supply chain. Hence, this field is constantly evolving and, in recent years, showed increasing interest in UASs. In fact, 19 papers among those reviewed focused on the use of drones in healthcare logistics.

The introduction of UAVs has proven to be a feasible and viable mean to support the healthcare logistic chain, especially when the facilities are under high-stress conditions and in dire need situations [54,55,69]. Particularly for low-income countries, drones emerged as a cost-effective method to improve efficiency and reduce logistics inadequacies and inequities by removing geographic barriers, increasing timeliness, and improving the accessibility of supplies, equipment, and remote care [54,69,71,72]. While all of these studies emphasize the cost savings that the implementation of drones could provide, in an economic evaluation proposed by Zailani et al. [62], the cost-effectiveness of drones was compared with ambulances for the transportation blood between the Sabah Women and Children Hospital and the Queen Elizabeth II Hospital on Borneo island. As a result, although drone transportation of blood products costs more per minute compared to the ambulance, the significantly shorter transport time of the drone offsets its cost.

The effective and reliable implementation of UAVs requires the continuous proposal of new solutions or updating and improvement of the underlying algorithms. For example, in a study by Shankar et al. [58], a platform that enables UAVs to transmit medical data effectively, ensuring reliable communication, was successfully developed. The framework also incorporates AI algorithms to enable intelligent decision-making processes for the UAV systems, allowing for the efficient real-time analysis of medical data collected through internet of medical things (IoMT) devices, leading to improved diagnosis and treatment options [58]. Also, a major part of the analyzed experimental research [56,57,60,61,65,73] deals with the solution and optimization of various routing and path planning problems, thus allowing for the whole drone system to minimize the travel time, reduce energy consumption, and improve overall efficiency.

However, it is not always possible to achieve both a high speed and minimal energy consumption using the same algorithm, as in the experimental study by Al-Rabiaah et al. [56]. Their case study in the Portland metropolitan area showed that the developed novel heuristic had a higher average energy consumption (around 30%) compared to the Gurobi, greedy, and 3SH algorithms, even though it achieved the best coverage in the shortest time. Therefore, several studies [59,64,66–68,70] propose the design of healthcare logistics systems that include the use of drones. Some describe a model designed to achieve faster delivery times [68,70], while others focused on proposing a complete drone service

framework, integrating all actors/stakeholders involved in the UAV service and outlining their interactions [64,66].

In a study aimed at designing a highly automated drone service for the delivery of pharmaceuticals at San Raffaele Hospital in Milan [66], a cocreation methodology was applied, and the needs of the main users were identified through semi-structured interviews and visualization material. The authors itemized the service requirements for the daily use of drones by a hospital to move biomedical material within its premises as a routine mode of transport.

However, almost all of the models proposed identified privacy [67] and safety issues [64], adverse weather conditions [59], and the need to have appropriate human resources [67] as limitations that need to be overcome.

### 3.3.4. Air Ambulance

Air ambulances have gained considerable recognition as a viable solution for enhancing access to remote or congested areas, thus contributing to improved emergency management.

In a retrospective analysis, Maddry et al. [75] analyzed the medical evacuation patient care records of USA military personnel injured in a military operation between 2011 and 2014. The findings indicated that utilizing air ambulances for eligible patients who did not receive a lifesaving intervention en route (approximately 50 percent) could have alleviated the burden on onsite medical personnel. This approach would have also led to reduced transportation times and combat-related morbidity and mortality.

The technical feasibility of utilizing flying cars for medical emergencies has been examined in a design study [74]. The simulation results emphasize the importance of increasing battery capacity to effectively deploy flying cars in emergencies. Additionally, the development of quieter rotors is crucial for environmental considerations. These observations are supported by the findings of Goyal and Cohen [76], who conducted a comprehensive feasibility study exploring the operational and market viability of the air ambulance industry. It is evident that the mentioned improvements would enhance the reliability and cost-effectiveness of air ambulances, although personnel requirements continue to be a primary driver of operational costs.

### 3.3.5. Other Medical Applications

As the consensus around the use of drones in healthcare is growing, new sophisticated applications have been developed in recent years.

Many drone medical applications could find their use in out-of-hospital settings. In fact, UAVs are used to detect mobility and consciousness from video images [81] or to evaluate breathing either by landing on victims' bodies or hovering over them [78].

An interesting implementation is suggested by Sheng et al. [77], who proposed a UAV-assisted autonomous drug delivery system for first aid. Their system consists of a drone, a contact-triggered microneedle applicator (CTMA), and a microneedle patch containing the emergency therapeutics. In vivo studies confirmed that this system successfully implemented autonomous first aid in a hypoglycemic minipig model, but further investigations need to be conducted for human applications. Drones have also been employed inside hospitals to provide healthcare services to patients; for example, they can help to assist patients in psychiatric long-term care by allowing them to fly the drone through eye movements in order to give them the opportunity to see the outside environment [80].

Moreover, UAVs are also suggested for telemedicine purposes, as in one experimental study [79], wherein drones enabled the medical professional to guide a subject to conduct a remotely mentored lung examination on himself, or in the theoretical model proposed in [82], in which UAVs resulted in being a flexible and cost-effective solution for the remote monitoring of the vital signs of patients and real-time scheduling of the transmission of vital signs.

### 3.3.6. Public Acceptance

Two separate research studies have examined the public perception and acceptance of drones in healthcare.

One study aimed at investigating the individual and institutional factors regarding the future role of drones in healthcare provision through a self-administered questionnaire addressed to 400 employees of three healthcare organizations in Norway [83]. The findings indicate a generally positive perception of drone usage across various professions, age groups, and locations. Additionally, factors such as working in an innovative environment, previous experience with technological changes in the workplace, and supportive leadership were identified as influential drivers shaping individual beliefs regarding the utilization of drones as innovative solutions in future healthcare services. Similarly, a questionnaire conducted in multiple hospitals in Malaysia sought to assess the opinion of rural healthcare workers towards drone delivery of medicines and vaccines [84]. In this case, slightly more than half of the participants expressed a positive attitude towards the use of drones for such purposes, thus highlighting the need for the further exploration of the perceived barriers and facilitators associated with the adoption of drone technology in this context.

### 3.3.7. Regulatory Framework

The only study among those reviewed focusing on the regulatory framework is a retrospective data analysis by Król-Całkowska and Walczak [85]. This research evaluated the provisions outlined in Polish and EU regulations regarding the permissibility of utilizing drones in airspace to protect public health aimed at identifying potential modifications to existing laws and demonstrates the feasibility of drone implementation for these purposes.

The main result that emerged from this study is the crucial necessity of establishing a comprehensive regulatory framework to ensure the secure and effective utilization of drones in healthcare. Currently, EU regulations offer a general framework for drone operations; however, individual Member States have the flexibility to establish supplementary rules and requirements. Specifically, in the case of Poland, the Polish Civil Aviation Authority (CAA) introduced specific regulations pertaining to drone usage in healthcare. These regulations include licensing requirements for drone operators, categorization of drone operations based on their nature, and limitations on flight altitudes and proximity to individuals and structures. Additionally, the Polish CAA has established protocols for acquiring flight permits and has implemented distinct regulations for operating drones in urban areas.

Overall, the authors stated the importance of comprehending and adhering to both EU regulations and country-specific legislation when deploying drones in healthcare, thus ensuring that operations are conducted in a secure, lawful, and compliant manner. Also, they highlighted the considerable potential of drones in mitigating health risks during the COVID-19 pandemic and beyond, while emphasizing the need for further research and collaboration to fully harness the capabilities of this emerging technology in healthcare settings.

## 4. Discussion

This scoping review has provided a comprehensive overview of the current state of research on the use of drones in the medical sector by capturing the body of work emerging in recent years. In continuity with the previous research, drones have emerged as a promising technological tool with the potential to revolutionize various aspects of healthcare provision.

The use of drones in the transport of biomedical goods, such as medicines, vaccines, and blood and biological samples, offers numerous advantages in terms of time savings and accessibility. The reviewed studies consistently demonstrate that drones can reduce delivery times, especially in remote or hard-to-reach areas [20–22,28]. Recent research efforts have elaborated on the possibility of timely administering even specialist treatment, e.g.,

naloxone for opioid overdose, arguing that drones would significantly reduce intervention times with respect to ambulances [86]. Notably, the successful transportation of human organs via drones represents a key milestone in the field [27]. Previous studies addressed the issue of the impact of drone vibrations on the integrity of the transported material [5]. While the reviewed articles indicate that the vibrations did not affect the quality of the transported items, further research is needed to assess the effects of drone vibrations on a wider range of biomedical goods [24,25]. Similarly, research on the effects of temperature and humidity is ongoing [26,32], with an effort to increase reliability by implementing an integrated system for real-time monitoring and both passive and active temperature control [26]. The authors of the present review argue that the transportation of biomedical materials via drones is intricately tied to compliance with dangerous goods regulations (DGR). A considerable number of substances intended for medical purposes fall under the purview of the "Technical Instructions for the Safe Transport of Dangerous Goods by Air" (Doc 9284), published by the ICAO [87]. A comprehensive audit by Grote et al. [33] provides a valuable examination of the regulations governing the transport of medical cargoes, shedding light on the complex landscape of compliance and safety considerations. The impact of dangerous goods regulations, as discussed by Grote and colleagues, extends to the very core of drone logistics for medical cargoes. These regulations introduce stringent requirements that can significantly influence the planning, execution, and viability of drone-based medical transport. To mitigate the risks associated with such operations, the European legislator has introduced the option to utilize crash-protected containers [88]. In this regard, the sole means for a UAV operator to circumvent the requirement of subjecting a drone to airworthiness certification is to demonstrate that, under the same conditions as the intended operations, the transported dangerous goods cannot pose a threat to bystanders or the environment in the event of accidents [88]. These measures underscore the need for new standards to which container manufacturers can adhere to demonstrate compliance. It would be highly beneficial for future research to focus on developing effective safety tests. In healthcare use cases, given the implications that DGR can have on logistics (e.g., the need for trained personnel, additional precautions in handling transported materials), considerations should also extend to the physical infrastructure for delivery. The analyzed articles did not address the topic of automated systems for loading and unloading transported materials, such as drone-in-a-box systems. However, the design of these systems cannot overlook compliance with such regulations. As a result, the importance of training healthcare personnel and all individuals involved in the delivery process becomes evident for the successful and safe execution of the entire operation.

Out-of-hospital cardiac arrest is a critical medical emergency in which timely defibrillation is crucial for survival. The use of AED-equipped drones has shown promise in reducing response times and increasing survival rates. While simulations and experiments consistently suggest the potential benefits of AEDs delivered by drones, real-world scenarios may introduce additional challenges, such as adverse weather conditions and obstacles [40]. Cost-effectiveness remains a topic of debate, with some studies suggesting AED drone systems can be more cost-efficient than traditional solutions, while others highlight the expenses associated with their implementation [38,45,52]. The feasibility and economic sustainability of AED drone deployment needs further investigation, with evidence suggesting the results may be highly influenced by factors related to different geographic and environmental contexts.

Optimization techniques and route planning algorithms have been a focus of research to minimize travel time and energy consumption. A variety of novel techniques have proved efficient at maximizing patient coverage [56], reducing the number of UAVs used and the total routing distance [57], and leading to the better distribution of resources [60]. However, balancing fast delivery time and energy efficiency remains a challenge [56,57,60,61,65,73]. This matter emerges as the foremost key challenge to reaching the full applicability of drones in healthcare logistics and especially in specific applications such as air ambulances.

Recent advances have shown promising results in simulation studies revolving around mission performance optimization [89,90].

Drones are being explored for various other medical applications, including assessing patient conditions [78,81] and telemedicine support [79,82]. These applications offer innovative solutions for remote healthcare, but further research and development are required to address technical and regulatory challenges.

Our scoping review did not yield papers concerning the potential risks associated with drones, including those posed to individuals on the ground and the potential for collisions with manned aircraft. It is essential to thoroughly assess these risks before engaging in drone operations, as such an assessment forms the foundation for determining the feasibility of these operations. Furthermore, the outcomes of the risk assessment typically lead to adjustments in the operational characteristics to ensure the reduction of risks to an acceptable level. In fact, on the basis of the results of the risk assessment, modifications may be made to the drone's flight route, impacting its performance parameters and, notably, its flight range and battery capacity. It is worth mentioning that recent research efforts have been conducted to estimate the ground risk based on a dynamic model [91]. Complementing similar models to estimate the ground risk to the more obviously dynamic nature of air risk in unmanned aircraft system (UAS), traffic management (UTM) systems could significantly contribute to lowering risks without hindering the operation's times.

The comprehensive integration of UAVs in healthcare demands a thorough understanding of the process to gain flight permissions, especially concerning beyond-visual-line-of-sight (BVLOS) operations. It is worth noting that, under the current European regulations, BVLOS delivery operations require the issuance of flight authorization by the CAA that is contingent on the results of the risk assessment. This results in the temporary reservation of the airspace volume required by the operation. In the UK and in Europe, UAV operators can be granted a Light UAS operator Certificate (LUC), which allows them to self-authorize their operations. As noted by Grote et al., however, this does not absolve LUC holders of the responsibility of gaining authorization for the transportation of dangerous goods, if such is the case [33]. In conclusion, how a regulatory framework can be effectively navigated in practice for medical networks linking hospitals and clinics needs careful consideration by highly experienced drone operators. Moreover, future research efforts could be made to effectively roll out UTM systems able to dynamically allocate temporarily reserved areas to manage air traffic at "very low levels".

### 4.1. Limitations of This Review

This scoping review adopted a systematic approach aimed at reducing biases. However, a few limitations must be considered. First, this review did not include grey literature. The choice of excluding nonpeer-reviewed literature ensured higher data quality and reliability, but it may result in the omission of potentially valuable and novel information not subjected to peer review. Moreover, the timespan which the current study focused on may be deemed limited, but it is coherent with the aim of this review, which is to offer a general understanding of the current challenges in drone research for healthcare to direct future research efforts on overcoming topical limitations.

### 4.2. Directions of Future Research

While the reviewed literature highlighted the significant potential of drones in healthcare, it is essential to recognize and address the challenges and limitations associated with their integration into the healthcare systems. These challenges encompass both technological and societal aspects and should be the focus of future research efforts.

Technological challenges:

1. Weather dependency: Many studies emphasize the sensitivity of drone operations to adverse weather conditions, such as strong winds, rainfall, or extreme temperatures. Weather can disrupt drone services causing their occasional unavailability, which is a significant concern for critical healthcare deliveries. UAV technical design should take

into account such environmental factors for a wider adoption of drone operations in healthcare on a daily basis.

2. Battery lifespan: The limited battery life of drones is a critical technological constraint. Long-distance deliveries may require scheduled stops for battery replacement or recharging, which makes this a challenge that encompasses not only the battery and the device itself but also infrastructure and urban planning considerations. Optimization techniques to drone control should be implemented for a more efficient use of battery lifespan; however, a variety of constraints that could intervene in real-life situations, such as obstacles and path planning in uncertain environment, should be taken into account when developing such systems.

3. Technical reliability: Ensuring the reliability of drone systems is paramount. Technical failures or malfunctions during transportation missions can have severe consequences, particularly when transporting life-saving medical supplies or organs, thus highlighting the importance of developing and adhering to technical standards.

4. Transport capacity: The payload capacity of drones can be limited, affecting the volume and the variety of medical goods that can be transported. Finding the right balance between payload capacity and drone size is essential.

5. Storage and temperature control: Maintaining temperature control and ensuring the stability of perishable goods during transport, such as blood samples or organs, is a technological challenge. Effective solutions for temperature control and maintaining the quality of medical items need to be developed.

Societal challenges:

1. Public acceptance: despite Generally positive perceptions, some individuals express concerns of fear regarding the use of drones in healthcare. In light of this review, a general feeling of unsafety around drones according to societal perceptions needs to be overcome before considering the deployment of medical use cases in urban areas.

2. Privacy and safety: Ensuring the privacy and safety of individuals during drone operations is a significant concern. Regulatory frameworks must include provisions for safeguarding privacy and addressing safety issues.

3. Regulatory gaps: The absence of comprehensive and standardized regulations tailored to drone delivery operations in healthcare is a persistent issue. Regulatory gaps need to be filled to provide clear guidelines for safe and lawful drone operations. In particular, technical standards to demonstrate the conformity to DGR intended for traditional aviation need to be adapted to UAV operations.

4. Human resources: Effective drone operations in healthcare require trained personnel. Ensuring adequate workforce with the skills to operate, maintain, and manage drones is essential.

5. Cost-effectiveness: The cost-effectiveness of drone-based healthcare services is a topic of debate. While some studies suggest cost savings, the initial expenses associated with drone implementation, including equipment and training, can be a barrier for adoption. A comprehensive review on this topic would aid in pinpointing the most cost-effective business cases, which can function as trailblazers for the less cost-effective ones, paving the way for more efficient and economical practices in the future.

6. Data security: managing and securing the data collected during UAV operations, especially in telemedicine, remote sensing, and remote monitoring applications, is crucial for protecting patient information and the safety of the operations.

7. Environmental impact: the environmental impact of drone operations, including noise pollution and wildlife disruption, needs to be considered and mitigated in advance.

In conclusion, while the use of drones in healthcare offers tremendous potential for improving patient care, addressing the technological and societal challenges and limitations is essential for successful implementation. Collaborative efforts among researchers, policymakers, healthcare professionals, and the public are essential to overcome these

hurdles and fully harness the benefits of drone technology in healthcare. Future research should continue to focus on developing innovative solutions and best practices to ensure safe, efficient, and accessible drone-based healthcare services.

**Author Contributions:** Conceptualization, S.D.S., P.J.C. and A.G.; methodology, S.D.S.; investigation, S.D.S., P.J.C. and A.G.; data curation, S.D.S., P.J.C., A.G. and S.M.; writing—original draft preparation, S.D.S., P.J.C., A.G. and S.M.; writing—review and editing, S.D.S., P.J.C., A.G. and S.M.; supervision, A.S.; funding acquisition, A.S. All authors have read and agreed to the published version of the manuscript.

**Funding:** This research was funded by the European Union H2020 Research and Innovation Programme under Grant Agreement No. 101006828—Flying Forward 2020.

**Data Availability Statement:** Data are contained within the article.

**Conflicts of Interest:** The authors declare no conflict of interest.

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
