# Peer review of "Challenges for the Routine Application of Drones in Healthcare: A Scoping Review"

_drones, doi:10.3390/drones7120685_

Round 1

Reviewer 1 Report

Comments and Suggestions for Authors

This is a timely review covering a very emotive topic. I would be keen to see this expanded considerably to bring out the key findings from the compiled papers and the subsequent gaps in knowledge and understanding which is lacking in several places. Key comments are:

-         The abstract largely describes what the review ‘will’ do and does not bring out any of the key findings and particularly, gaps in knowledge. This needs addressing.

-         The definition of UAV needs to be edited to ‘uncrewed aerial vehicle’ throughout

-         P.5, L189 ‘However, it is worth noting that more than 23% of the participants expressed fear, while 34.5% felt nervous about the use of drones.’ To really add value, the review needs to bring out the important detail from these papers. What ‘fears’ were expressed? Were these related to reliability of the service or whether the characteristics of drone flight may result in adverse impacts on the quality of the cargoes carried?

-         Again, P.5. L196, ‘Their usage has been recognized as a feasible [21], [23] and cost-effective solution [20] in healthcare’ … there is so much to discuss from this one line but no details are given. The debate about whether drones are cost effective is contentious as the knowledge about forward and reverse medical supply chains and the real logistics requirements is not well understood. Much more detail is needed from these references … what were the case studies, circumstances etc?

-         P.6, L198 ‘Moreover, through the persistent optimization of algorithms focusing on routing and path planning issues, drone systems have achieved higher efficiency and decreased energy consumptions [36], [30], [29], [86]’ ….. but how many of these papers focussed on reality commercial systems? The vast majority are theoretical studies (apart from the Ziplines of this world). Much more detail is needed from these references.

-         The review is missing a few key areas that need addressing:

i)                    There is also scope for the movement of specialist treatments by drone …. E.g. Noloxone for the treatment of overdose patients (see the work by Dr Paul Royall at Kings College)

ii)                   There needs to be discussion on the issue of dangerous goods classified medical cargoes and the impacts on drone logistics (see the work by Matt Grote, University of Southampton …. https://www.bing.com/ck/a?!&&p=dea95ac2647fc07cJmltdHM9MTY5Nzg0NjQwMCZpZ3VpZD0xMTU3ODM3Mi01YmI0LTYwNjAtMzc2My05MGMzNWE4YzYxZTQmaW5zaWQ9NTM1MQ&ptn=3&hsh=3&fclid=11578372-5bb4-6060-3763-90c35a8c61e4&psq=matt+grote+dangerous+goods+regulations&u=a1aHR0cHM6Ly93d3cubWRwaS5jb20vam91cm5hbC9kcm9uZXMvc3BlY2lhbF9pc3N1ZXMvbWVkaWNpbmU&ntb=1 ). This also involves the new Crash Proof Container regulations which have huge implications for the viability of drone logistics for medical cargoes gong forward. This is not mentioned at all.

iii)                 There needs to be explicit mention of the air and ground risk posed by drones (and in relation to ii) above, (the issues of weather tolerance and inclement conditions impacting on flight reliability are discussed)). The potential impacts on routing and the implications on range and battery performance need covering (e.g.       Drones | Free Full-Text | Ground Risk Assessment for Unmanned Aircraft Systems Based on Dynamic Model (mdpi.com); publications | Aliaksei Pilko)

iv)                 The issues of how medical practitioners actually interact with UAVs (in terms of the consignor and consignee) needs covering. E.g. Inteliport, drone in a box etc given health and safety legislation.

v)                   The whole issue of gaining flight permissions in terms of flying beyond-visual-line-of-sight and how this is practically done for medical networks between hospitals and clinics needs coverage (temporary danger areas etc)

These gaps in knowledge and the barriers to the wider adoption of UAVs in healthcare need to be brought out for this review to make a meaningful contribution.

Comments on the Quality of English Language

Needs a good editing by a native English speaker

Reviewer 2 Report

Comments and Suggestions for Authors

-              The article presents an interesting methodological development, the area of knowledge is still developing, which addresses the topic from the review of existing literature, this contributes to the identification of gaps in specific topics or aspects of what has been studied, applied and investigated. until today.

-              The only weakness found has to do with table 1, which lacks a column that describes the ideas and knowledge expressed in the cited sources, because as presented it does not contribute nor is it clear in the interpretation of the results and contribution of the sources. cited.

-              Each section of the article, from the introduction, materials and methods, analysis of results to references, are well developed methodologically and evidence the fulfillment of the objectives of the work presented.

-              Each of the themes or topics analyzed is addressed in detail, citing updated sources from high-impact publications such as Scopus and Web of Science.

-              The presentation of sources in the references section is clear and corresponds to the style used for citations and references with precise and clear information that leads to a practical search for each reference.

 This article develops an interesting topic, makes an  important analysis of literature riview and its evolution in research.

Reviewer 3 Report

Comments and Suggestions for Authors

Round 2

Reviewer 1 Report

Comments and Suggestions for Authors

The authors have addressed the comments raised. I still feel that the overall term of 'uncrewed aerial vehicles' should be used (highighting the other core terms that are also used i.e. unmanned)

Comments on the Quality of English Language

Minor final proof read needed

Author Response

The authors appreciate the reviewer's effort to improve the quality of the manuscript. After careful consideration, the authors decided to use "UAV" throughout the paper, and replaced the terms "UA" and "UAS" were necessary.

Moreover, a final proof read was performed as recommended, which resulted in minor editing of English language.